# ZeroI2V: Zero-Cost Adaptation of Pre-trained Transformers from Image to Video

## Abstract

Adapting image models to video domain is becoming an efficient paradigm for solving video recognition tasks. Due to the huge number of parameters and effective transferability of image models, performing full fine-tuning is less efficient and even unnecessary. Thus, recent research is shifting its focus towards parameter-efficient image-to-video adaptation. However, these adaptation strategies inevitably introduce extra computational cost to deal with the domain gap and temporal modeling in videos. In this paper, our goal is to present a zero-cost adaptation paradigm (ZeroI2V) to transfer the image transformers to video recognition tasks (i.e., introduce zero extra cost to the adapted models during inference). To achieve this goal, we present two core designs. First, to capture the dynamics in videos and reduce the difficulty of achieving image-to-video adaptation, we exploit the flexibility of self-attention and introduce the spatial-temporal dual-headed attention (STDHA) that efficiently endow the image transformers with temporal modeling capability at zero extra parameters and computation. Second, to handle the domain gap between images and videos, we propose a linear adaption strategy which utilizes lightweight densely placed linear adapters to fully transfer the frozen image models to video recognition. Due to its customized linear design, all newly added adapters could be easily merged with the original modules through structural reparameterization after training, thus achieving zero extra cost during inference. Extensive experiments on five widely-used video recognition benchmarks show that our ZeroI2V can match or even outperform previous state-of-the-art methods while enjoying superior parameter and inference efficiency.

## 1 Introduction

Adapting pre-trained foundation models such as BERT (Devlin et al., 2019) and GPT (Radford et al., 2018; 2019; Brown et al., 2020) through efficient strategies has yielded excellent performance on downstream tasks in natural language understanding. This new paradigm is becoming popular in computer vision due to the available pre-trained image models such as CLIP (Radford et al., 2021) and DINO (Caron et al., 2021; Oquab et al., 2023). These models could be easily adapted to downstream tasks through linear probe, fine-tuning or even zero-shot recognition, exhibiting robustness and strong transfer capabilities similar to those of large-scale language models. Recently, *parameter-efficient transfer learning* (PETL) (Jia et al., 2022; Chen et al., 2022; Nie et al., 2022; Lin et al., 2022b; Pan et al., 2022) is becoming an efficient paradigm to adapt these large pre-trained models due to their huge numbers of parameters and high computational cost of full fine-tuning.

For video understanding, there exist several large pre-trained video models (Tong et al., 2022; Wang et al., 2023) from self-supervised learning, but these models are of high computational complexity due to the joint spatiotemporal attentions. Therefore, adapting pre-trained image models to video domain through efficient strategies is still a practical solution to video recognition. In fact, the state-of-the-art video networks have long relied on the pre-trained image models by inflating the kernels (Carreira & Zisserman, 2017; Liu et al., 2022a; Arnab et al., 2021; Liu et al., 2022c) or inserting plug-and-play temporal modules (Wang et al., 2016; Lin et al., 2022a; Li et al., 2020; Liu et al., 2021b; Wang et al., 2021a). However, most of these methods necessitate full fine-tuning, which involves updating all the model parameters during training on video datasets. As the scale of pre-trained models increases, full fine-tuning becomes impractical due to the high training costs and the risk of overfitting or even catastrophic forgetting when the downstream data is limited. In addition, these methods often inevitably introduce extra cost to the adapted video models due to these newly added modules.

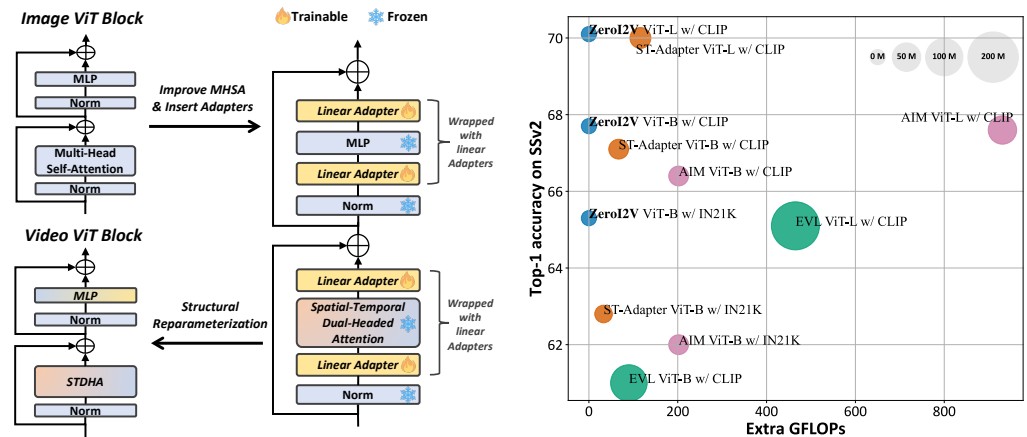

Figure 1: **Left**: Our zero-cost image-to-video transfer learning method. **Right**: Comparison of PETL mehods on SSv2 validation set. For more intuitive comparison, the views of the methods in the figure are all $8 \times 3 \times 1$. Two core techniques enable us to achieve superior performance on video tasks without introducing additional computation and parameters during inference.

In this paper, we aim to present a new efficient paradigm of adapting image transformers to video downstream tasks with two main objectives. First, inspired by the PETL methods in NLP (Houlsby et al., 2019; Lester et al., 2021; Hu et al., 2022; Li & Liang, 2021) and image understanding (Jia et al., 2022; Chen et al., 2022; Nie et al., 2022), we aim to devise a parameter-efficient transfer technique from image to video, which can effectively reduce the risk of over-fitting and greatly improve the training efficiency. Second, to overcome the issue of high computation in the adapted video models, we try to present a zero-cost adaptation method without introducing any extra computations to the final video models during inference. This zero-cost adaptation would allow for more efficient deployment of transferred video models in the real applications.

To achieve the above two objectives, we propose a novel zero-cost transfer learning method (as shown in Figure 1) that can utilize the off-the-shelf pre-trained image transformers to *achieve excellent performance on video tasks without additional parameters and computation during inference*. To be specific, for the temporal modeling required for video tasks, we transform multi-head self-attention into *spatio-temporal dual-head attention* (STDHA) by reassigning some heads to achieve temporal modeling at zero computation and zero parameters. For image-to-video transfer, we explore the strategy of using linear adapters to fully adapt the parameters of each part of the model and merge them with the frozen original parameters through structural reparameterization after training, thus achieving zero-cost during inference.

To summarize, we make the following contributions: **1)** We propose a new approach for *parameter-efficient image-to-video transfer learning* that can achieve the zero-cost adaptation of transformers from image to video without introducing additional computation and parameters during inference. **2)** We introduce a novel attention mechanism named *Spatial-Temporal Dual-Headed Attention* (STDHA), which utilizes the flexibility of self-attention to achieve temporal modeling without introducing extra computation and parameters. **3)** To the best of our knowledge, we are the first to investigate the achievement of zero-cost image-to-video adaptation through the utilization of a linear structure. And we establish a benchmark in this field by conducting extensive experiments with a diverse range of adaptation strategies. **4)** Our method achieves comparable or even better performance than state-of-the-art methods on five action recognition benchmarks while enjoying the advantage of parameter and inference efficiency.

## 2  RELATED WORK

**Pre-trained image transformers**  The powerful scalability of ViT (Dosovitskiy et al., 2021) brings more possibilities to the pre-trained image model. In addition to the traditional supervised approach (Dosovitskiy et al., 2021; Zhai et al., 2022; Liu et al., 2022b), recent works (He et al., 2020; Bao et al., 2022; Caron et al., 2021; He et al., 2022a; Oquab et al., 2023) utilize self-supervised learning to effectively learn representations from unlabeled data. Moreover, several works (Rad-

ford et al., 2021; Li et al., 2022a; Tschannen et al., 2022; Cherti et al., 2022) adopt large-scale multi-modal data (*e.g.*, text-image pairs) to learn visual representations with great transferability. Our proposed adaptation strategy can leverage these off-the-shelf pre-trained image transformers to achieve outstanding performance on video tasks.

**Video action recognition**  State-of-the-art methods for video action recognition have long relied on image models. Previous works for action recognition can be classified into two categories: one is to extend the image model for spatial-temporal modeling by inflating weights and structures (Carreira & Zisserman, 2017; Feichtenhofer et al., 2019; Feichtenhofer, 2020; Liu et al., 2022c; Li et al., 2022b; Fan et al., 2021a; Li et al., 2022c), while the other is to directly utilize the image model as the backbone and insert plug-and-play modules for temporal modeling (Wang et al., 2016; Zhou et al., 2018; Lin et al., 2022a; Liu et al., 2021b; Wang et al., 2021a). Following the success of new training paradigms in image understanding, several works have attempted to learn transferable video representations via self-supervised learning (Tong et al., 2022; Wang et al., 2022; Lu et al., 2023; Wang et al., 2023) or multi-modal video-text pre-training (Li & Wang, 2020; Wang et al., 2021b; Ni et al., 2022; Li et al., 2023). However, the above methods usually require full fine-tuning of the entire model or training from scratch, resulting in high training costs and additional computational overhead. In this work, we avoid the above problems by adapting the pre-trained image transformers to video tasks in an efficient manner.

**Parameter-efficient transfer learning**  To address the issue of training inefficiency caused by the continuous growth of model size, Parameter-efficient transfer learning (PETL) is initially introduced in NLP  (Houlsby et al., 2019; Pfeiffer et al., 2020; 2021; Lester et al., 2021; Li & Liang, 2021; Hu et al., 2022; Zaken et al., 2022) and subsequently applied to vision tasks (Jia et al., 2022; Lian et al., 2022; He et al., 2022b; Chen et al., 2022; Nie et al., 2022). These techniques aim to achieve comparable or even superior performance on other tasks by fine-tuning only a small subset of trainable parameters. Most PETL methods (Jia et al., 2022; Chen et al., 2022; He et al., 2022b; Lian et al., 2022; Zhang et al., 2022; Nie et al., 2022) in vision domain are limited to transfer within the same modality (*e.g.*, image-to-image or video-to-video). In contrast, our research focuses on image-to-video transfer learning. Despite progress made by recent studies (Lin et al., 2022b; Pan et al., 2022; Yang et al., 2023), these methods require additional computation and parameters for temporal modeling of video tasks and image-to-video adaptation. For example, EVL (Lin et al., 2022b) incorporates an additional temporal transformer decoder, while ST-Adapter (Pan et al., 2022) introduces additional adapters with depth-wise 3D convolution layers. Similarly, AIM (Yang et al., 2023) adds extra adapters and necessitates an additional time attention calculation at each block. In contrast to previous works, our proposed method eschews the introduction of additional computation or parameters during inference, yet still achieves comparable or superior performance compared to previous methods.

## 3  METHODOLOGY

In this section, we first briefly revisit the basic block of ViT (Sec. 3.1), and then discuss how to utilize the flexibility of self-attention to achieve temporal modeling without introducing additional computation and parameters (Sec. 3.2). Finally, we explain how we implement zero-cost image-to-video adaptation with a serial linear structure (Sec. 3.3).

### 3.1  PRELIMINARY

The original ViT (Dosovitskiy et al., 2021) block consists of two types of network layers: multi-head self-attention (MHSA) and multi-layer perceptron (MLP). As shown in Figure 1, a ViT block consists of MHSA and MLP connected in series in a residual structure:

$$z_l = x_l + \text{MHSA}(\text{LN}(x_l)), \tag{1}$$

$$x_{l+1} = z_l + \text{MLP}(\text{LN}(z_l)), \tag{2}$$

where LN denotes layer normalization (Ba et al., 2016) and $x_l$ represents the input to the $l$-th ViT block. We review their specific implementation details. For the sake of simplicity, we ignore the bias and denote $X \in \mathbb{R}^{n \times d}$ as input of MHSA and MLP.

MHSA first performs three different linear projections $W_{\text{attn}}^Q, W_{\text{attn}}^K, W_{\text{attn}}^V \in \mathbb{R}^{d \times d}$ on the input $X$ to obtain the query $Q$ and key-value pairs $K, V$. These are then evenly divided into $h$ heads by channel.

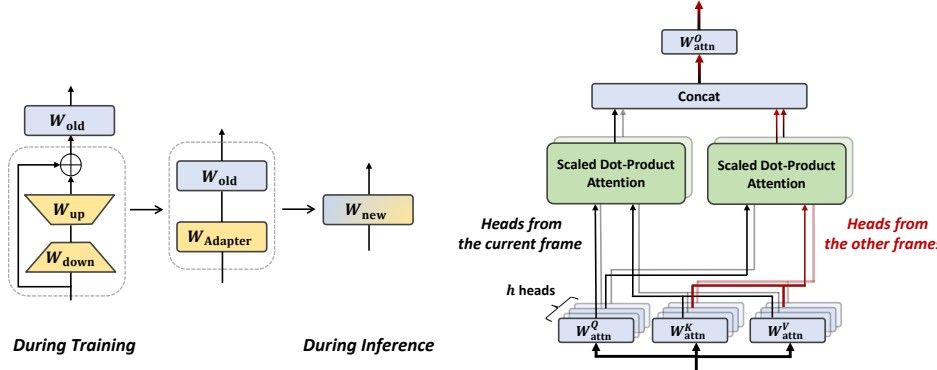

(a) Layer merging via structural reparameterization   (b) Spatial-temporal dual-headed attention

Figure 2: **Illustration of the proposed linear adaptation and STDHA.**

Each head independently performs the scaled dot-product attention calculation. Finally, the heads are concatenated by channel and then a linear projection $W_{\text{attn}}^O \in \mathbb{R}^{d \times d}$ is performed to obtain the final calculation result:

$$Q, K, V = XW_{\text{attn}}^Q, XW_{\text{attn}}^K, XW_{\text{attn}}^V, \tag{3}$$

$$\text{head}_i = \text{Attention}(Q_i, K_i, V_i), \tag{4}$$

$$\text{MHSA}(X) = \text{Concat}(\text{head}_1, \cdots, \text{head}_h)W_{\text{attn}}^O. \tag{5}$$

MLP involves two linear projections $W_{\text{mlp}}^{\text{up}} \in \mathbb{R}^{d \times d'}, W_{\text{mlp}}^{\text{down}} \in \mathbb{R}^{d' \times d}, d' > d$ and one non-linear activation function $\sigma$:

$$\text{MLP}(X) = \sigma(XW_{\text{mlp}}^{\text{up}})W_{\text{mlp}}^{\text{down}}. \tag{6}$$

### 3.2   Zero-Cost Temporal Modeling

Applying image models to video tasks often requires the incorporation of additional modules for temporal modeling, which not only introduces additional parameters and computation, but also results in additional training costs. In this work, we address temporal modeling from three key perspectives: (1) Capability of capturing the temporal dynamics. (2) Reducing the difficulty of image-to-video adaptation. (3) Minimizing the introduction of additional computation and parameters compared to the original model. Michel et al. (2019) suggests that most heads are redundant given the rest of the model. Inspired by this, we attempt to reassign some heads as *temporal heads* in the multi-head attention to perform temporal modeling tasks, while the remaining heads continue to perform their original spatial modeling tasks as *spatial heads*, thereby achieving efficient spatial-temporal modeling.

**Spatial-temporal dual-headed attention (STDHA)**   As shown in Figure 2b, consider an input sequence $X = \{x_1, x_2, \cdots, x_T\}$ where $x_t \in \mathbb{R}^{n \times d}$. Let the query and key-value pairs obtained after linear projection of the $x_t$ be $Q^t, K^t, V^t \in \mathbb{R}^{n \times d}$. We divide the $h$ heads of the MHSA into two groups of size $h - k$ and $k$. One group of heads queries the key-value pairs at the current time $t$ to perform *spatial modeling*, while the other group of heads queries the key-value pairs at other times $t + \Delta t_i$ to perform *temporal modeling*. Finally, the information from the two groups of heads is aggregated by a linear projection to perform *spatial-temporal modeling*:

$$\text{S-head}_i = \text{Attention}(Q_i^t, K_i^t, V_i^t), \tag{7}$$

$$\text{T-head}_i = \text{Attention}(Q_i^t, K_i^{t+\Delta t_i}, V_i^{t+\Delta t_i})(\Delta t_i \neq 0), \tag{8}$$

$$\text{STDHA}(X) = \text{Concat}(\text{T-head}_1, \cdots, \text{T-head}_k, \text{S-head}_{k+1} \cdots \text{S-head}_h)W_{\text{attn}}^O, \tag{9}$$

where $\Delta t_i$ represents the time offset of the key-value pair of the $i$-th head. We did not directly use temporal attention or temporal convolution for the temporal modeling like previous works (Lin et al., 2022b; Pan et al., 2022; Yang et al., 2023). Instead, we design a more efficient spatiotemporal modeling operator by decoupling spatial modeling and temporal modeling to different heads:

- For the spatial head, it still only needs to complete the spatial modeling task as the original image transformer, which reduces the difficulty of achieving image-to-video adaptation.

- For the temporal head, it actually implements the inter-frame attention mechanism with frames at different times. Zhang et al. (2023) have demonstrated the effectiveness of inter-frame attention mechanism for modeling motion information, which is crucial for action recognition tasks. In addition, as shown in Table 1c, we can achieve both short-distance and long-distance modeling by controlling the $\Delta t_i$ of the temporal head, which enables us to achieve enhanced temporal modeling capabilities.

**Comparison with other zero-cost operators**    There have been several previous attempts (Bulat et al., 2021; Zhang et al., 2021; Xiang et al., 2022) to use image transformers to achieve efficient temporal modeling at zero parameters and zero computation. For example, Bulat et al. (2021) achieves approximations to full space-time attention by mixing tokens from adjacent frames. Zhang et al. (2021) performs temporal modeling by using channel shift on the cls tokens of different frames. Xiang et al. (2022) mixes information from adjacent frames using temporal patch shift and temporal channel shift before MHSA. However, these methods do not take advantage of the inherent characteristics of the transformer structure. By decoupling the learning of spatial and temporal information with head relocation, STDHA maintains the ***purity of key-value pair information*** within the same head, thereby achieving better spatial-temporal information learning than other zero-cost temporal modules. And STDHA ***simultaneously captures both short-range and long-range dependencies***, rather than being limited to adjacent frames. As shown in the Table 1, these two key distinctions enable our STDHA to achieve superior spatial-temporal modeling.

### 3.3 ZERO-COST IMAGE-TO-VIDEO ADAPTATION

Inspired by LoRA (Hu et al., 2022), we can fine-tune the model using a linear structure and then merge it with the original model during inference. However, to deal with the domain gap between images and videos, previous works (Lin et al., 2022b; Yang et al., 2023; Pan et al., 2022) often use nonlinear structures to achieve stronger transfer capabilities. Therefore, we need to further consider how to achieve effective image-to-video transfer using only a linear structure.

**Layer merging via structural reparameterization**    Let $W_{\text{old}}$ represent the frozen weights of the original model, and $W_{\text{new}}$ represent the new trainable weights. Reviewing the structure of LoRA, it uses a low-rank decomposition matrix $W_{\text{LoRA}}$ parallel to the original weights:

$$W_{\text{new}} = W_{\text{LoRA}} + W_{\text{old}} = W_{\text{up}}W_{\text{down}} + W_{\text{old}}. \tag{10}$$

In this work, we use a serial linear structure called ***Linear Adapter*** to fine-tune the original parameters. As shown in Figure 2a, we use structural reparameterization to perform layer merging after training:

$$W_{\text{new}} = W_{\text{Adapter}}W_{\text{old}} = (I + W_{\text{up}}W_{\text{down}})W_{\text{old}}, \tag{11}$$

where $I$ is the identity matrix, $W_{\text{up}} \in \mathbb{R}^{m \times k}, W_{\text{down}} \in \mathbb{R}^{k \times n}$, bottleneck width $k \ll \min(m, n)$. As seen in Table 2, compared to parallel structures, serial structures can be more flexibly inserted into the network structure (*e.g.*, for non-square matrices, under the same bottleneck dimension, using LoRA requires a larger number of parameters compared to Linear Adapter), which endows it with better transfer capabilities.

**Full adaptation with densely placed linear adapters**    By observing the structure of MHSA and MLP, we can see that all their trainable parameters concentrate on the linear projections at both ends of the structure. Therefore, fine-tuning the model essentially updates these linear projections. Previous works (Yang et al., 2023; Pan et al., 2022) often selectively tune part of the parameters (*e.g.*, placing only an adapter before MHSA) instead of tuning all parameters to avoid excessive additional computational and parameter costs, while we can achieve zero-cost ***full adaptation*** by tuning all parameters through wrapping MHSA and MLP with linear adapters. Table 2 shows that full adaptation enables us to achieve excellent image-to-video transfer performance with a linear structure, compensating for the performance degradation caused by the removal of nonlinearity.

Table 1: **Ablation study on STDHA.** Most of the symbols in the table have been declared in the methodology section 3. (a)$R_c$ denotes channel change ratio, "Shift' refers to temporal channel shift, while "HR" denotes head relocation as used by STDHA. (b) We use a multiset to represent the time offsets of different heads (*e.g.*, "$1 \cdot 2$" means that there are 2 heads with $\Delta t = 1$). When $\Delta t$=0, it represents a spatial head. (c) "Temporal RF" refers to temporal receptive field of a single STDHA.

(a) Comparison of temporal modeling methods,

| $R_c$ | Method | Top-1 |
|---|---|---|
| 1/6 | [cls] token shift | 61.4 |
| | Shift $QKV$ | 64.5 |
| | Shift $KV$ | 64.6 |
| | HR $QKV$ | 64.8 |
| | HR $KV$ (STDHA) | **66.0** |
| 1/4 | Shift KV | 64.0 |
| | HR $KV$ (STDHA) | 65.8 |

(b) Effect of the number of temporal heads

| Backbone | $\Delta t$ of heads | $k$ | Top-1 |
|---|---|---|---|
| ViT-B ($h$=12) | $\{1 \cdot \frac{1}{2}, -1 \cdot \frac{1}{2}, 0 \cdot 11\}$ | 1 | 64.8 |
| | $\{1 \cdot 1, -1 \cdot 1, 0 \cdot 10\}$ | 2 | **66.0** |
| | $\{1 \cdot 2, -1 \cdot 2, 0 \cdot 8\}$ | 4 | 65.6 |
| | $\{1 \cdot 3, -1 \cdot 3, 0 \cdot 6\}$ | 6 | 65.6 |
| ViT-L ($h$=16) | $\{1 \cdot 1, -1 \cdot 1, 0 \cdot 14\}$ | 2 | 67.7 |
| | $\{1 \cdot 2, -1 \cdot 2, 0 \cdot 12\}$ | 4 | **68.5** |
| | $\{1 \cdot 3, -1 \cdot 3, 0 \cdot 10\}$ | 6 | 68.3 |

(c) Effect of temporal receptive field at at different input lengths.

| Frames | $\Delta t$ of heads | Temporal RF | Top-1 |
|---|---|---|---|
| 8 | $\{1 \cdot 1, 0 \cdot 11\}$ | 2 | 64.7 |
| | $\{1 \cdot 1, -1 \cdot 1, 0 \cdot 10\}$ | 3 | **66.0** |
| | $\{1 \cdot 1, -1 \cdot 1, 2 \cdot 1, 0 \cdot 9\}$ | 4 | 65.5 |
| | $\{1 \cdot 1, -1 \cdot 1, 2 \cdot 1, -2 \cdot 1, 0 \cdot 8\}$ | 5 | 65.7 |
| 16 | $\{1 \cdot 1, -1 \cdot 1, 0 \cdot 10\}$ | 3 | 67.2 |
| | $\{1 \cdot 1, -1 \cdot 1, 2 \cdot 1, 0 \cdot 9\}$ | 4 | 67.3 |
| | $\{1 \cdot 1, -1 \cdot 1, 2 \cdot 1, -2 \cdot 1, 0 \cdot 8\}$ | 5 | **67.8** |
| | $\{1 \cdot 1, -1 \cdot 1, 2 \cdot 1, -2 \cdot 1, 3 \cdot 1, 0 \cdot 7\}$ | 6 | 67.6 |
| | $\{1 \cdot 1, -1 \cdot 1, 2 \cdot 1, -2 \cdot 1, 3 \cdot 1, -3 \cdot 1, 0 \cdot 6\}$ | 7 | 67.3 |
| 32 | $\{1 \cdot 1, -1 \cdot 1, 0 \cdot 10\}$ | 3 | 67.3 |
| | $\{1 \cdot 1, -1 \cdot 1, 2 \cdot 1, 0 \cdot 9\}$ | 4 | 67.8 |
| | $\{1 \cdot 1, -1 \cdot 1, 2 \cdot 1, -2 \cdot 1, 0 \cdot 8\}$ | 5 | 68.5 |
| | $\{1 \cdot 1, -1 \cdot 1, 2 \cdot 1, -2 \cdot 1, 3 \cdot 1, 0 \cdot 7\}$ | 6 | **68.6** |
| | $\{1 \cdot 1, -1 \cdot 1, 2 \cdot 1, -2 \cdot 1, 3 \cdot 1, -3 \cdot 1, 0 \cdot 6\}$ | 7 | 68.4 |
| | $\{1 \cdot 1, -1 \cdot 1, 2 \cdot 1, -2 \cdot 1, 3 \cdot 1, -3 \cdot 1, 4 \cdot 1, 0 \cdot 5\}$ | 8 | 68.2 |

# 4 EXPERIMENTS

## 4.1 EXPERIMENTS SETUP

We evaluate our method on five widely-used video recognition benchmarks: two large-scale datasets, namely Kinetics-400 (K400) (Carreira & Zisserman, 2017) and Something-Something V2 (SSv2) (Goyal et al., 2017), in addition to three smaller-scale datasets, UCF101 (Soomro et al., 2012), HMDB51 (Kuehne et al., 2011) and Diving48 Li et al. (2018). We also evaluate our method on action detection dataset AVA (Gu et al., 2018), and the results can be found in the Appendix B. This diverse dataset selection allows for a comprehensive evaluation of our model across various scales and domains. The specific model configuration and training strategy also can be found in the Appendix A. For most main experiments, we use ViT-B and ViT-L pre-trained by CLIP (Radford et al., 2021) as our backbone models, and the additional results using other backbone architectures and pretrained weights can be found in the Appendix B.

## 4.2 ABLATION STUDY

To validate the effectiveness of our method on image-to-video transfer and temporal modeling, we first conduct ablation experiments on the SSv2 dataset. All ablation experiments were performed using ViT-B/16 with 8 input frames unless specified.

**Effectiveness of STDHA**  Table 1a compares STDHA with other zero-cost temporal modeling methods. The [cls] token shift is implemented according to the original paper (Zhang et al., 2021), with [cls] token shift performed before MHSA and MLP. The temporal channel shift operation refers to TPS (Xiang et al., 2022), which shifts a portion of the channels for each head. It can be seen that

Table 2: **Comparison of adaption strategies.** "Width" refers to the bottleneck width of LoRA/Adapter. "Tunable Params" refers to extra trainable parameters besides the parameters of the ViT backbone and linear classifier. "✓" and "✗" indicate whether the corresponding weights have undergone fine-tuning, and "✓" indicates that $W_{attn}^Q$, $W_{attn}^K$ and $W_{attn}^V$ share the same adapter. "Latency" refers the latency of inference with 3 test samples. All results are obtained using a same V100-32G with PyTorch-builtin mixed precision.

| Method | Weights of ViT block | | | | | | Tunable Params (M) | Bottleneck Width | Latency (ms) | SSv2 Top-1 |
|---|---|---|---|---|---|---|---|---|---|---|
| | $W_{attn}^Q$ | $W_{attn}^K$ | $W_{attn}^V$ | $W_{attn}^O$ | $W_{mlp}^{up}$ | $W_{mlp}^{down}$ | | | | |
| Full Fine-tuning | ✓ | ✓ | ✓ | ✓ | ✓ | ✓ | 86 | - | 28.9 | 63.2 |
| Linear Probe | ✗ | ✗ | ✗ | ✗ | ✗ | ✗ | 0 | - | 28.9 | 20.0 |
| Only tuning temporal head | ✓ | ✓ | ✓ | ✓ | ✗ | ✗ | 4.6 | - | 28.9 | 59.6 |
| ST-Adapter (Pan et al., 2022) | ✓ | ✓ | ✓ | ✓ | ✓ | ✓ | 14 | 192 | 41.0 | 66.2 |
| | ✓ | ✓ | ✓ | ✗ | ✓ | ✗ | 14 | 384 | 38.8 | 65.8 |
| LoRA (Hu et al., 2022) | ✓ | ✗ | ✓ | ✗ | ✗ | ✗ | 7 | 192 | | 64.2 |
| | ✓ | ✓ | ✓ | ✗ | ✗ | ✗ | 14 | 192 | | 65.0 |
| | ✓ | ✗ | ✓ | ✗ | ✓ | ✓ | 25 | 192 | 28.9 | 64.3 |
| | ✓ | ✗ | ✓ | ✗ | ✓ | ✓ | 17 | 128 | | 65.6 |
| | ✓ | ✓ | ✓ | ✓ | ✓ | ✓ | 32 | 192 | | 65.0 |
| | ✓ | ✓ | ✓ | ✓ | ✓ | ✓ | 21 | 128 | | 65.5 |
| Adapter w/ GELU | ✓ | ✓ | ✓ | ✓ | ✓ | ✓ | 7 | 96 | 37.3 | 65.6 |
| | ✓ | ✓ | ✓ | ✗ | ✓ | ✗ | 7 | 192 | 34.9 | 64.6 |
| | ✓ | ✓ | ✓ | ✓ | ✓ | ✗ | 10 | 192 | 36.3 | 66.1 |
| | ✓ | ✓ | ✓ | ✓ | ✓ | ✓ | 14 | 192 | 38.4 | 66.1 |
| Linear Adapter (Ours) | ✓ | ✓ | ✓ | ✓ | ✓ | ✓ | 7 | 96 | | 65.0 |
| | ✓ | ✓ | ✓ | ✗ | ✓ | ✗ | 7 | 192 | | 64.4 |
| | ✓ | ✓ | ✓ | ✓ | ✓ | ✗ | 10 | 192 | 28.9 | 65.2 |
| | ✓ | ✓ | ✓ | ✓ | ✓ | ✓ | 14 | 192 | | 66.0 |
| | ✓ | ✓ | ✓ | ✓ | ✓ | ✓ | 20 | 192 | | **66.3** |
| | ✓ | ✓ | ✓ | ✓ | ✓ | ✓ | 14 | 128 | | 66.2 |

STDHA significantly outperforms other methods at the same channel change ratio, demonstrating the importance of preserving the purity of information within each head.

**Effect of the number of temporal heads and temporal receptive field** We examined the influence of the number of temporal heads and the temporal receptive field in ViT-B and ViT-L. Our findings, detailed in Tables 1b and 1c, suggest that the optimal proportion of temporal heads in ViT lies between 1/6 and 1/4. Regarding the temporal receptive field, our results indicate that for 8-frame inputs, a field of 3 is sufficient, while for longer inputs (16 or 32 frames), performance improves with an increase in the field from 3, saturating at around 5 or 6. Consequently, we employ different STDHA configurations based on input length.

**Comparison of adaptation strategies** In Table 2, we compare the image-to-video transfer ability of our method with a diverse range of commonly used adaptation methods. For a fair comparison, we all use STDHA with the same setting to provide temporal modeling capabilities. From the results, we can observe that:

- Even with only a small number of parameters being fine-tuned, our linear adapter tuning significantly outperforms full fine-tuning (66.3 vs 63.2). Despite updating the fewest parameters, linear probe performs poorly in image-to-video transfer.

- Even if we only tune the temporal head, we can still achieve about 95% of the full fine-tuning performance. This suggests that extensive fine-tuning of the spatial head may not be necessary to attain satisfactory transfer performance due to the decoupling of spatial and temporal modeling reduces the difficulty of adaptation.

- Our *Full Adaptation* strategy is not only effective for linear adapters, but also for non-linear adapters such as the ST-Adapter and GELU Adapter. It not only enhances their adaptation performance, but also eliminates the performance gap between linear and non-linear structures.

Table 3: **Results on Kinetics-400 validation set**. Views = #frames $\times$ #spatial crops $\times$ #temporal clips. "GFLOPs" means $10^9$ FLOPs, "M" means $10^6$. "Extra GLOPs" refers to the extra computation added to the original ViT under the same number of views. "New Params" refers to additional parameters during inference besides the parameters of the original ViT backbone and linear classifier.

| Methods | Pretrain | Views | GFLOPs | Extra GFLOPs | Param(M) | New Param(M) | Top-1 | Top-5 |
|---|---|---|---|---|---|---|---|---|
| *Methods with full fine-tuning* | | | | | | | | |
| UniFormer-B (Li et al., 2022b) | IN1K | 32×3×4 | 3108 | - | 50 | - | 83.0 | 95.4 |
| TimeSformer-L (Bertasius et al., 2021) | IN21K | 96×3×1 | 7140 | - | 121 | - | 80.7 | 94.7 |
| VideoSwin-L (Liu et al., 2022c) | IN21K | 32×3×4 | 7248 | - | 197 | - | 83.1 | 95.9 |
| MViTv2-L(↑312) (Li et al., 2022c) | IN21K | 40×5×3 | 42420 | - | 218 | - | 86.1 | 97.0 |
| ViViT-L/16x2 FE (Arnab et al., 2021) | JFT | 32×3×1 | 11940 | - | 311 | - | 83.5 | 94.3 |
| MTV-L (Yan et al., 2022) | JFT | 32×3×4 | 18050 | - | 876 | - | 84.3 | 96.3 |
| ViT-B/16 (Pan et al., 2022) | CLIP | 8×1×3 | 422 | 0 | 86 | 0 | 81.0 | 95.5 |
| ActionCLIP-B/16 (Wang et al., 2021b) | CLIP | 32×3×10 | 16893 | 13 | 142 | 56 | 83.8 | 97.1 |
| X-CLIP ViT-L/14 (Ni et al., 2022) | CLIP | 8×3×4 | 7896 | 107 | 420 | 116 | 87.1 | 97.6 |
| Text4Vis ViT-L/14 (Wu et al., 2023) | CLIP | 32×3×4 | 19944 | - | 347 | 43 | 87.1 | 97.4 |
| *Methods with PETL* | | | | | | | | |
| VideoPrompt ViT-B/16 (Ju et al., 2022) | CLIP | 16×5×1 | - | - | - | - | 76.9 | 93.5 |
| ST-Adapter ViT-B/16 (Pan et al., 2022) | CLIP | 32×1×3 | 1821 | 133 | 93 | 7 | 82.7 | 96.2 |
| EVL ViT-L/14 (Lin et al., 2022b) | CLIP | 8×1×3 | 2022 | 76 | 362 | 58 | 86.3 | - |
| AIM ViT-L/14 (Yang et al., 2023) | CLIP | 32×1×3 | 11208 | 3425 | 341 | 38 | **87.5** | **97.7** |
| **ZeroI2V** ViT-B/16 | CLIP | 8×1×3 | 422 | 0 | 86 | 0 | 83.0 | 95.8 |
| **ZeroI2V** ViT-B/16 | CLIP | 16×1×3 | 844 | 0 | 86 | 0 | 83.4 | 96.2 |
| **ZeroI2V** ViT-B/16 | CLIP | 32×1×3 | 1688 | 0 | 86 | 0 | 83.7 | 96.4 |
| **ZeroI2V** ViT-L/14 | CLIP | 8×1×3 | 1946 | 0 | 304 | 0 | 86.3 | 97.4 |
| **ZeroI2V** ViT-L/14 | CLIP | 16×1×3 | 3892 | 0 | 304 | 0 | 86.8 | 97.6 |
| **ZeroI2V** ViT-L/14 | CLIP | 32×1×3 | 7783 | 0 | 304 | 0 | 87.2 | 97.6 |

- Due to the inflexibility of the parallel structure, for non-square matrices like $W_{\mathrm{mlp}}$, LoRA requires more parameters under the same bottleneck width. It needs to decrease the bottleneck width of the low-rank matrix to align it with the number of parameters of the linear adapter. However, this reduction in bottleneck width can limit its adaptation ability, ultimately leading to results that are significantly worse than those of the Linear Adapter.

### 4.3 COMPARISONS WITH THE STATE OF THE ART

**Results on K400**  As shown in Table 3, our method has significant advantages over traditional full fine-tuning methods, achieving better performance with much lower computational cost. For example, our ZeroI2V ViT-L/14 with an input of 8 frames outperforms MViTv2 (Li et al., 2022c) (86.3 vs 86.1), while requiring more than 20 times fewer GFLOPs (1946 vs 42420). Compared to multi-modal methods such as ActionCLIP (Wang et al., 2021b) and X-CLIP (Ni et al., 2022), which require an additional text branch and fine-tune the entire model end-to-end, our ZeroI2V can achieve comparable performance using only the visual encoder. Moreover, although our proposed ZeroI2V doesn't increase computational or parameter costs during inference compared with the previous PETL method, it can still achieve similar or even better performance. For example, on ViT-B/16, ZeroI2V with an input of 8 frames can surpass ST-Adapter (Pan et al., 2022) with an input of 32 frames (83.0 vs 82.7) with much lower GFLOPs (422 vs 1821). On ViT-L/14, ZeroI2V achieves the same performance as EVL (Lin et al., 2022b), which requires an additional 58M parameters. And ZeroI2V achieves comparable performance to AIM (Yang et al., 2023) (87.2 vs 87.5) with a nearly 30% reduction in GFLOPs (7783 vs 11208).

**Results on SSv2**  As shown in Table 4, thanks to the effectiveness of STDHA in temporal modeling, our method outperforms most full fine-tuning methods, even though many of them have been pre-trained on the Kinetics dataset. And our ZeroI2V has a significant improvement compared to directly full fine-tuning ViT-L/16 pre-trained with CLIP (70.1 vs 48.7) with the same number of parameters and computation. Compared to other PETL methods, ZeroI2V outperforms ST-Adapter (Pan et al., 2022) on ViT-B/16 (70.1 vs 69.5) with lower GFLOPs (1688 vs 1955). Additionally, ZeroI2V significantly surpasses both EVL (Lin et al., 2022b) and AIM (Yang et al., 2023) (71.4 vs 66.7, 70.6) on ViT-L/14 with much lower GFLOPs (3892 vs 9641, 11508) and new parameters (0M vs 175M, 50M).

Table 4: **Results on Something-Something v2 validation set.** K400†/K600†indicates that the model is pre-trained on both IN21K (except for Uniformer (Li et al., 2022b) which uses IN1K) and K400/K600. The other notations are the same as Table 3.

| Methods | Pretrain | Views | GFLOPs | Extra GFLOPs | Param(M) | New Param(M) | Top-1 | Top-5 |
|---|---|---|---|---|---|---|---|---|
| *Methods with full fine-tuning* | | | | | | | | |
| TimeSformer-L (Bertasius et al., 2021) | IN21K | 64×3×1 | 7140 | - | 121 | - | 62.4 | - |
| ViViT-L (Arnab et al., 2021) | K400† | 16×3×4 | 11892 | - | 311 | - | 65.4 | 89.8 |
| MTV-B(↑320) (Yan et al., 2022) | K400† | 32×3×4 | 11160 | - | 310 | - | 68.5 | 90.4 |
| VideoSwin-B (Liu et al., 2022c) | K400† | 32×3×1 | 963 | - | 89 | - | 69.6 | 92.7 |
| MViTv2-L(↑312) (Li et al., 2022c) | K400† | 40×3×1 | 8484 | - | 213 | - | **73.3** | **94.1** |
| UniFormer-B (Li et al., 2022b) | K600† | 32×3×1 | 777 | - | 50 | - | 71.2 | 92.8 |
| ViT-L/14 (Dosovitskiy et al., 2021) | CLIP | 8×3×1 | 1946 | 0 | 86 | 0 | 48.7 | 77.5 |
| ILA ViT-L/14 (Tu et al., 2023) | CLIP | 8×3×4 | 10884 | 3100 | 529 | 225 | 67.8 | 90.5 |
| *Methods with PETL* | | | | | | | | |
| ST-Adapter ViT-B/16 (Pan et al., 2022) | CLIP | 32×3×1 | 1955 | 267 | 100 | 14 | 69.5 | 92.6 |
| EVL ViT-L/14 (Lin et al., 2022b) | CLIP | 32×3×1 | 9641 | 1858 | 479 | 175 | 66.7 | - |
| AIM ViT-L/14 (Yang et al., 2023) | CLIP | 32×3×1 | 11508 | 3725 | 354 | 50 | 70.6 | 92.7 |
| **ZeroI2V** ViT-B/16 | CLIP | 8×3×1 | 422 | 0 | 86 | 0 | 67.7 | 90.8 |
| **ZeroI2V** ViT-B/16 | CLIP | 16×3×1 | 844 | 0 | 86 | 0 | 69.4 | 91.7 |
| **ZeroI2V** ViT-B/16 | CLIP | 32×3×1 | 1688 | 0 | 86 | 0 | 70.1 | 92.4 |
| **ZeroI2V** ViT-L/14 | CLIP | 8×3×1 | 1946 | 0 | 304 | 0 | 70.1 | 91.8 |
| **ZeroI2V** ViT-L/14 | CLIP | 16×3×1 | 3892 | 0 | 304 | 0 | 71.4 | 93.0 |
| **ZeroI2V** ViT-L/14 | CLIP | 32×3×1 | 7783 | 0 | 304 | 0 | **72.2** | **93.0** |

Table 5: **Inference latency and throughput.** All results are obtained using a same V100-32G with PyTorch-builtin mixed precision, using a batch size of 1 to measure latency and the optimal possible batch size to measure throughput before running out of memory.

| Model | Views | GFLOPs | Latency (ms) | Throughput (V/s) | K400 Top-1 | SSv2 Top-1 |
|---|---|---|---|---|---|---|
| Uniformer-B (Li et al., 2022b) | 32×4 | 1036 | 245.38 | 4.24 | 82.9 | - |
| EVL ViT-B/16 (Lin et al., 2022b) | 8×3 | 454 | 53.87 | 24.04 | 82.9 | 61.0 |
| ViT-B/16 (Dosovitskiy et al., 2021) | 8×3 | 422 | 28.72 | 40.08 | 81.0 | 44.0 |
| **ZeroI2V** ViT-B/16 | 8×3 | 422 | 28.89 | **40.08** | **83.0** | **67.7** |

**Results on smaller datasets**  On three relatively small datasets, our method achieves top-1 accuracies of **98.6%**, **83.4%** and **91.4%** on UCF101, HMDB51 and Diving48, respectively. This demonstrates a clear performance advantage over both full-finetuning methods and PETL methods previously. A detailed comparison could be found in the Table 10, 11 of Appendix B

**Comparison of inference efficiency**  We compared the inference efficiency of our method with other methods on the same hardware device. As shown in Table 5, under comparable accuracy, the throughput of our method is 10 times that of Uniformer (Li et al., 2022b), Compared to the original ViT-B, our method introduces negligible additional latency during inference while achieving superior performance. In comparison with EVL (Lin et al., 2022b), it can also be seen that the impact of the additional computational module on the actual runtime latency (28.89 ms vs 53.87 ms) is greater than that reflected by GFLOPs (422 vs 454).

## 5  CONCLUSIONS

In this work, we have presented a new zero-cost approach for *parameter-efficient image-to-video transfer learning*, called ZeroI2V. By fully leveraging the powerful representational capabilities of pre-trained image models, our approach enables image transformers to perform video tasks without introducing extra cost during inferences. Our proposed STDHA achieves efficient spatial-temporal modeling at zero extra computation and parameters. In addition, through structural reparameterization and full adaptation strategies, we have successfully attempted to use a linear structure to achieve zero-cost image-to-video adaptation for the first time. ZeroI2V shows strong performance compared to previous full fine-tuning and PETL methods on five widely-used action recognition benchmarks while maintaining advantages in parameter and inference efficiency. Due to the simplicity and versatility of our method, we believe it can be easily extended to other video tasks and even multi-modal understanding tasks. We will further investigate this direction in future work.

**Reproducibility**  To ensure all the results can be reproduced, we give the details of the model and training hyperparameters in our experiments (see Appendix A). And we will publish our source code after the review process.

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

In this appendix, we provide more details of ZeroI2V from the following aspects:

- Implementation details of our method are in § A.
- Experimental results with other pre-trained weights and backbone architectures, as well as training cost analysis, can be found in § B.
- Visualization of our proposed Spatial-Temporal Dual-Headed Attention (STDHA) is in § C.
- Limitations and societal impact are in § D
- License of the datasets and pre-trained models are in § E

## A  IMPLEMENTATION DETAILS OF OUR METHOD

### A.1  MODEL DETAILS

Table 6: **Model details.** We use a multiset to represent the time offsets of different heads (*e.g.*, "$1 \cdot 2$" means that there are 2 heads with $\Delta t = 1$). When $\Delta t = 0$, it represents a spatial head."Temporal RF" refers to temporal receptive field of a single STDHA. "Num. Adapters" refers to the number of linear adapters per ViT block.

(a) Model details for Kinetics400.

| Backbone | Frames | $\Delta t$ of heads | Temporal RF | Num. Adapters |
|---|---|---|---|---|
| ViT-B ($h$=12) | 8 | $\{1 \cdot 1, -1 \cdot 1, 0 \cdot 10\}$ | 3 | 4 |
| | 16 | $\{1 \cdot 1, -1 \cdot 1, 2 \cdot 1, 0 \cdot 9\}$ | 4 | 4 |
| | 32 | $\{1 \cdot 1, -1 \cdot 1, 2 \cdot 1, -2 \cdot 1, 3 \cdot 1, 0 \cdot 7\}$ | 6 | 4 |
| ViT-L ($h$=16) | 8 | $\{1 \cdot 2, -1 \cdot 2, 0 \cdot 12\}$ | 3 | 4 |
| | 16 | $\{1 \cdot 2, -1 \cdot 2, 2 \cdot 1, 0 \cdot 11\}$ | 4 | 4 |
| | 32 | $\{1 \cdot 2, -1 \cdot 2, 2 \cdot 1, -2 \cdot 1, 3 \cdot 1, 0 \cdot 9\}$ | 6 | 4 |

(b) Model details for Something-Something v2.

| Backbone | Frames | $\Delta t$ of heads | Temporal RF | Num. Adapters |
|---|---|---|---|---|
| ViT-B ($h$=12) | 8 | $\{1 \cdot 1, -1 \cdot 1, 0 \cdot 10\}$ | 3 | 6 |
| | 16 | $\{1 \cdot 1, -1 \cdot 1, 2 \cdot 1, -2 \cdot 1, 0 \cdot 8\}$ | 5 | 4 |
| | 32 | $\{1 \cdot 1, -1 \cdot 1, 2 \cdot 1, -2 \cdot 1, 3 \cdot 1, 0 \cdot 7\}$ | 6 | 4 |
| ViT-L ($h$=16) | 8 | $\{1 \cdot 2, -1 \cdot 2, 0 \cdot 12\}$ | 3 | 4 |
| | 16 | $\{1 \cdot 2, -1 \cdot 2, 2 \cdot 2, -2 \cdot 2, 0 \cdot 8\}$ | 5 | 4 |
| | 32 | $\{1 \cdot 2, -1 \cdot 2, 2 \cdot 1, -2 \cdot 1, 3 \cdot 1, 0 \cdot 9\}$ | 6 | 4 |
| Swin-B ($h = 4, 8, 16, 32$) | 32 | Stage 1: $\{1 \cdot 1, 0 \cdot 3\}$
Stage 2: $\{1 \cdot 1, -1 \cdot 1, 0 \cdot 6\}$
Stage 3: $\{1 \cdot 1, -1 \cdot 1, 2 \cdot 1, -2 \cdot 1, 0 \cdot 12\}$
Stage 4: $\{1 \cdot 2, -1 \cdot 2, 2 \cdot 2, -2 \cdot 2, 0 \cdot 24\}$ | 2
3
5
5 | 4 |

Our model details are shown in the Table 6a and Table 6b. Due to different requirements for spatial modeling and temporal modeling in different datasets, there are slight differences in the specific implementation settings.

**Settings of STDHA**  For the settings of STDHA, we allocate 1/6 to 1/4 of the heads for temporal modeling based on the conclusions obtained from previous ablation experiments. For long inputs, we increase the absolute value of $\Delta t$ to obtain a larger temporal receptive field. When using Swin-B as the backbone, due to its four stages and different numbers of heads in each stage, we simply allocate temporal heads to each stage at a ratio of 1/4. Since its input length is halved after patch embedding (from 32 frames to 16 frames), we set the value of $\Delta t$ according to the best temporal receptive field of 16 frames, which is 5. Please note that we have not tried other configurations due to time constraints, so there may be better configurations.

**Number of adapters**  Considering the balance between performance and training cost, we only assign different adapters for each weight for the minimum setting (ViT-B with 8-frame input) on the SSv2 dataset, which requires 6 adapters. For all other settings, we only use 4 adapters. And the bottleneck ratios of all adapters are set to 0.25.

## A.2  TRAINING DETAILS

Table 7: **Training details of our method.**

| dataset | K400 | SSv2 |
|---|---|---|
| *Optimization settings* | | |
| optimizer | AdamW, learning rate=3e-4, weight decay=5e-2 | |
| batch size | 64 | |
| training epochs | 40 | 50 |
| *Sampling settings* | | |
| crop size | 224 | |
| frame sampling rate | 16 (for 8-frame input)
8 (for 16-frame input)
4 (for 32-frame input) | uniformly sample as TSN (Wang et al., 2016) |
| num. testing views | 3 temporal × 1 spatial | 1 temporal × 3 spatial |
| *Data augmentation settings* | | |
| RandAugment (Cubuk et al., 2020) | m=7, n=4 | |
| flip | 0.5 | |
| Random erasing (Zhong et al., 2020) | - | 0.25 |
| label smoothing | - | 0.1 |

As shown in the Table 7, our training strategy is similar to the previous methods (Pan et al., 2022; Yang et al., 2023). Considering that SSv2 requires stronger temporal modeling ability, we used a stronger data augmentation strategy following Liu et al. (2022c). In addition, for the full finetuing experiment using Swin-B as the backbone (Swin-B with STDHA), we use exactly the same training strategy as video swin transformer (Liu et al., 2022c).

**Implementation details of adaptation strategies**  The training configurations used for all the adaptation strategies are summarized as follows:

- For the comparison experiment of full finetuning ViT with CLIP pretrained, we use 1/10 of the learning rate to avoid training collapse.

- For the comparison experiment of only tuning temporal head, we froze the parameters related to the spatial head (only training the part of the parameters related to the temporal head, in other words, we only trained $W_{\text{attn}}^{Q^t}, W_{\text{attn}}^{K^t}, W_{\text{attn}}^{V^t} \in \mathbb{R}^{d \times d^t}, W_{\text{attn}}^{O^t} \in \mathbb{R}^{d^t \times d}$, where $d^t$ is the number of channels of the temporal head).

For any settings not explicitly mentioned, we assume they align with the training settings of the Linear Adapter.

## B  ADDITIONAL EXPERIMENTAL RESULTS

**Effect of bottleneck ratio**  We set the bottleneck dimension equal to the product of the ViT width and the bottleneck ratio. As shown in Table 8, a bottleneck ratio of 0.25 achieves a good trade-off between performance and the number of parameters. Therefore, we choose 0.25 as the bottleneck ratio for all subsequent experiments.

Table 8: **Effect of bottleneck ratio of linear adapters.**

| Ratio | Tunable Param(M) | Top-1 |
|---|---|---|
| 0.0625 | 3 | 64.2 |
| 0.125 | 7 | 65.0 |
| 0.25 | 14 | **66.0** |
| 0.5 | 28 | 65.8 |

**Experiments on action detection**  In addition to the task of action recognition, to understand the capability of our method in fine-grained spatial understanding, we also evaluate our method on action detection dataset AVA (Gu et al., 2018). Following the setting of VideoMAE (Tong

Table 9: **Comparing the state-of-the-art action detection methods on AVA 2.2.**

| Method | Pretrain | Frames | mAP |
|---|---|---|---|
| SlowFast-R101 (Feichtenhofer et al., 2019) | K400 | 8 | 23.8 |
| MViTv2-B (Li et al., 2022c) | K400 | 32 | 28.1 |
| VideoMAE-B (Tong et al., 2022) | K400 | 16 | 31.8 |
| VideoMAE-B (Tong et al., 2022) | K400 wo/ labels | 16 | 26.7 |
| CLIP ViT-B/16 | CLIP | 8 | 18.3 |
| **ZeroI2V** ViT-B/16 | CLIP | 8 | 26.4 |

et al., 2022), we evaluate the top 60 common classes using the mean Average Precision (mAP) as the metric under an IoU threshold of 0.5. As shown in Table 9, compared to using the original image CLIP features, our ZeroI2V achieved a significant performance improvement (26.4 vs 18.3) with the same number of parameters and computation. It's noteworthy that our method was not pre-trained on action recognition datasets such as Kinetics. Instead, we directly applied image-to-video transfer on the AVA dataset. Remarkably, our method still managed to achieve performance on par with full-finetuning methods and self-supervised methods that underwent pre-training using the Kinetics dataset, even when using only 8 frames as input. In summary, our ZeroI2V demonstrates outstanding potential in video tasks beyond recognition.

Table 10: **Comparing the state-of-the-art video recognition methods on UCF101 and HMDB51.** We test our method and report the 3-split mean Top-1 accuracy for both datasets following ST-Adapter (Pan et al., 2022).

| Method | Pretrain | UCF101 | HMDB51 |
|---|---|---|---|
| *Methods with full fine-tuning* | | | |
| STC (Diba et al., 2018) | K400 | 95.8 | 72.6 |
| ECO (Zolfaghari et al., 2018) | K400 | 93.6 | 68.4 |
| I3D (Carreira & Zisserman, 2017) | ImageNet+K400 | 95.6 | 74.8 |
| S3D (Xie et al., 2018) | ImageNet+K400 | 96.8 | 75.9 |
| SlowOnly-8x8-R101 (Feichtenhofer et al., 2019) | Kinetics+OmniSource | 97.3 | 79.0 |
| VideoPrompt (Ju et al., 2022) | CLIP | 93.6 | 66.4 |
| *Methods with PETL* | | | |
| ST-Adapter ViT-B/16 (Pan et al., 2022) | CLIP+K400 | 96.4 | 77.7 |
| ST-Adapter ViT-L/14 (Pan et al., 2022) | CLIP+K400 | 98.1 | 81.7 |
| **ZeroI2V** ViT-B/16 | CLIP | 95.6 | 73.7 |
| **ZeroI2V** ViT-B/16 | CLIP+K400 | 97.7 | 78.5 |
| **ZeroI2V** ViT-L/14 | CLIP | 97.8 | 79.9 |
| **ZeroI2V** ViT-L/14 | CLIP+K400 | **98.6** | **83.4** |

Table 11: **Comparing the state-of-the-art video recognition methods on Diving48.** We test our method with 1 temporal clip following AIM Yang et al. (2023).

| Method | Pretrain | Top-1 |
|---|---|---|
| *Methods with full fine-tuning* | | |
| TimeSformer-L (Bertasius et al., 2021) | IN21K | 81.0 |
| VideoSwin-B (Liu et al., 2022c) | IN21K | 81.9 |
| BEVT (Wang et al., 2022) | IN21K+K400 | 87.2 |
| SIFAR-B  (Fan et al., 2021b) | IN21K | 87.3 |
| *Methods with PETL* | | |
| AIM ViT-B/16 (Yang et al., 2023) | CLIP | 88.9 |
| AIM ViT-L/14 (Yang et al., 2023) | CLIP | 90.6 |
| **ZeroI2V** ViT-B/16 | CLIP | 89.7 |
| **ZeroI2V** ViT-L/14 | CLIP | **91.4** |

Table 12: **Results on K400 and SSv2 validation set with ImageNet21K pretrained.** Views = #frames × #spatial crops × #temporal clips. "GFLOPs" means $10^9$ FLOPs, "M" means $10^6$. "Extra GLOPs" refers to the extra computation added to the original ViT under the same number of views. "New Params" refers to additional parameters during inference besides the parameters of the original ViT backbone and linear classifier. Views for all methods are 8×1×3 for K400 and 8×3×1 for SSv2

| Methods | Pretrain | GFLOPs | Extra GFLOPs | Param(M) | New Param(M) | K400 Top-1 | SSv2 Top-1 |
|---|---|---|---|---|---|---|---|
| *Methods with full fine-tuning* | | | | | | | |
| TimeSformer (Bertasius et al., 2021) | IN21K | 590 | - | 121 | - | 78.0 | 59.5 |
| X-ViT (Bulat et al., 2021) | IN21K | 425 | - | 92 | - | 78.5 | 64.4 |
| *Methods with PETL & ViT-B/16* | | | | | | | |
| EVL (Lin et al., 2022b) | IN21K | 454 | 32 | 115 | 29 | 75.4 | - |
| ST-Adapter (Pan et al., 2022) | IN21K | 455 | 33 | 93 | 7 | 76.6 | 62.8 |
| AIM (Yang et al., 2023) | IN21K | 624 | 202 | 100 | 14 | **78.8** | 62.0 |
| **ZeroI2V** | IN21K | 422 | 0 | 86 | 0 | 78.6 | **65.3** |

**Experiments with ImageNet21K pre-trained weights** In order to investigate the adaptability of our method to different pre-trained weights, we conducted experiments using the same model and training settings on ImageNet21K pre-trained weights. The results are shown in Table 12. It can be seen that our method is still very effective under ImageNet21K weights and can surpass previous full fine-tuning methods. Compared to other PETL methods, our method shows stronger robustness. As shown in Figure 1, when using ImageNet21K pre-trained weights, the advantage of our method over other PETL methods is even greater than when using CLIP pre-trained weights. For example, when using CLIP weights, our method slightly surpasses ST-Adapter (Pan et al., 2022) (67.7 vs 67.1), while when using ImageNet21K weights, we have a clear advantage (65.3 vs 62.8).

Table 13: **Results on SSv2 validation set with Swin-B backbone.** K400†indicates that the model is pre-trained on both IN21K and K400. The other notations are the same as Table 12
.

| Methods | Pretrain | Views | GFLOPs | Param(M) | Tunable Param(M) | Top-1 | Top-5 |
|---|---|---|---|---|---|---|---|
| VideoSwin-B (Liu et al., 2022c) | K400† | 32×3×1 | 963 | 89 | 89 | 69.6 | **92.7** |
| PST-B (Xiang et al., 2022) | IN21K | 32×3×1 | 741 | 89 | 89 | 67.4 | 90.9 |
| SIFAR-B (Fan et al., 2021b) | IN21K | 32×3×1 | 789 | 87 | 87 | 62.6 | 88.5 |
| Swin-B w/ **STDHA** | IN21K | 32×3×1 | 741 | 89 | 89 | **70.0** | 92.1 |
| **ZeroI2V** Swin-B | IN21K | 32×3×1 | 741 | 89 | 14 | 67.8 | 91.4 |

**Experiments with other backbone** In order to verify the universality of our method, we conducted experiments using the Swin Transformer (Liu et al., 2021a) in Table 13, which has a hierarchical structure and local window attention. As shown in Table 1, although our method is not specifically designed and adjusted for it, it can still achieve performance comparable or even better than other full fine-tuning methods. To our surprise, when we used a full fine-tuning strategy to train Swin-B using STDHA, we achieved a top-1 accuracy of **70%**, which even surpassed VideoSwin-B (Liu et al., 2022c) pre-trained on the K400 dataset. From this, we can see that our designed STHDA is not only versatile but also has powerful temporal modeling capabilities. In addition, for backbones like Swin Transformer that have more inductive bias and have not been trained on large-scale image-text datasets, full fine-tuning may be able to better utilize the temporal modeling capabilities of STDHA.

**Comparison of training cost** We compared the training cost of our method with previous methods in Table 14. It can be seen that compared to previous full fine-tuning methods such as Uniformer (Li et al., 2022b) and ActionCLIP (Wang et al., 2021b), our method significantly reduces training cost. Compared to the previous PETL method, our method does not have a significant advantage in training efficiency due to the use of dense adapters. EVL (Lin et al., 2022b), which does not need to insert adapters into the frozen backbone, avoids some of the cost of backpropagation and therefore has lower memory usage. ST-Adapter (Pan et al., 2022), due to its fewer trainable parameters, has a

faster convergence speed, but its memory usage is close to our method. Nonetheless, in contrast to AIM (Yang et al., 2023) that impose an additional computational burden for temporal modeling, our STDHA method, which does not introduce extra learnable parameters, ensures that ZeroI2V maintains superior training efficiency. We believe that it is worthwhile and acceptable to exchange some training cost for a reduction in inference cost. We will also try to further reduce training cost by improving training strategies and adaptation strategies in the future.

Table 14: **Comparison of training cost**. Our results are obtained using a same V100-32G with PyTorch-builtin mixed precision, following EVL (Lin et al., 2022b). "†" indicates that the epoch is estimated based on the batch size and training steps of the original paper. "Memory" refers to the GPU memory usage when the batch size is 8.

| Model (Frames) | Dataset | Training Epochs | Training GPU Hours | Tunable Param (M) | Memory (G) | Top-1 |
|---|---|---|---|---|---|---|
| Uniformer-B (Li et al., 2022b) (32) | K400 | 110 | 5000 × V100 | 50 | - | 82.9 |
| ActionCLIP ViT-B/16 (Wang et al., 2021b) (16) | K400 | 50 | 480 × RTX3090 | 142 | - | 82.6 |
| EVL ViT-B/16 (Lin et al., 2022b) (8) | K400 | 53† | 60 × V100 | 29 | 2.2 | 82.9 |
|  | SSv2 | 46† | 75 × V100 | 98 | 5.6 | 61.0 |
| ST-Adapter ViT-B/16 (Pan et al., 2022) (8) | K400 | 11† | 23 × V100 | 7 | 6.9 | 82.0 |
|  | SSv2 | 38† | 60 × V100 | 14 | 7.6 | 67.1 |
| AIM ViT-B/16 (Yang et al., 2023) (8) | K400 | 30 | 120 × V100 | 11 | 8.7 | 83.9 |
|  | SSv2 | 50 | 150 × V100 | 14 | 9.0 | 66.4 |
| **ZeroI2V** ViT-B/16 (8) | K400 | 40 | 100 × V100 | 14 | 7.6 | 83.0 |
|  | SSv2 | 50 | 90 × V100 | 14 | 7.6 | 67.3 |

## C   VISUALIZATION

The motivation behind the design of STDHA is to enable simultaneous spatial and temporal modeling in an independent way before information fusion (ie. decoupling spatial and temporal modeling). In order to more intuitively demonstrate the temporal modeling capabilities of our proposed STDHA, we visualized the attention map of the last layer of the network. As shown in Figure 3, note that we visualize the attention map of the last transformer layer in our figure. Due to the temporal receptive field increases with the network depth, both the spatial head and the temporal head of this last layer have a global temporal receptive field about the input frames. But we can still observe that the spatial heads pay more attention to the information of the current frame while the temporal head pays more attention to the information of other frames. We compare the attention maps of STDHA and CLIP, and it can be seen that STDHA pays more attention to the interaction of objects in the video (such as the touch between hands and cups or Rubik's cubes), while CLIP only focuses on individual objects and does not capture the spatio-temporal dynamics in the video well.

## D   LIMITATIONS AND SOCIETAL IMPACT

**Limitations**   Our method has the following two main limitations:

- Although our method is very efficient during inference, the densely inserted linear adapters still need to participate in gradient calculation during training, which brings a non-negligible training cost. This makes our method still have a certain disadvantage in training cost compared to methods that use CLIP as an independent feature extractor (such as EVL (Lin et al., 2022b)). In the future, we need to consider more efficient training strategies and improve the structure of linear adapters to address this issue.

- Although STDHA has demonstrated powerful temporal modeling capabilities, it still requires consideration of the original number of heads in ViT and manual design of a head relocation strategy. Despite the ablation experiment results showing that our method's performance is relatively stable across different head relocation strategies, achieving better results still necessitates some manual design. Obtaining optimal head relocation strategies through manual design is obviously challenging. In future work, we aim to investigate methods for automatically designing head relocation strategies.

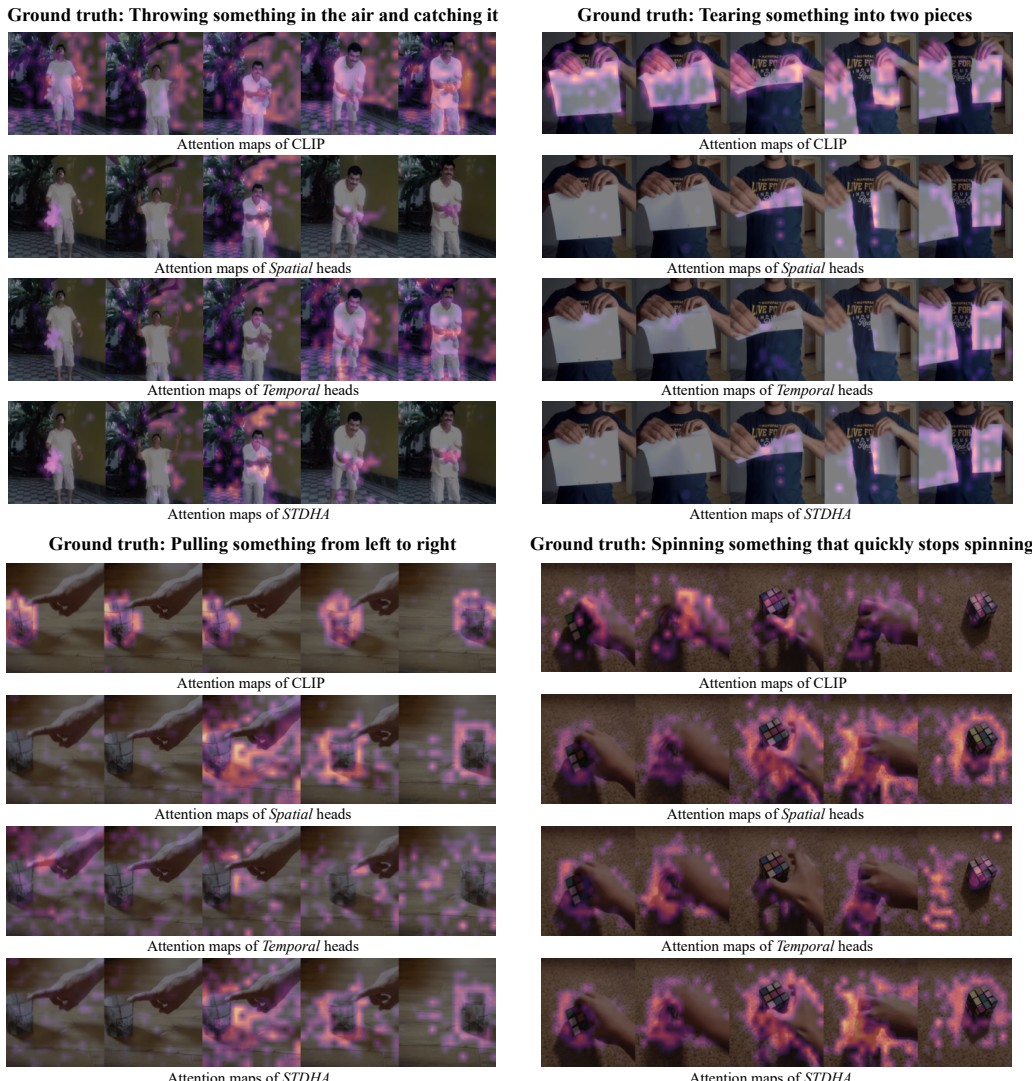

Figure 3: **Visualization of attention maps of CLIP, spatial heads, temporal heads and STDHA at the last layer generated by Grad-CAM (Selvaraju et al., 2020) on SSv2 validation set.**

**Societal impact**  Our ZeroI2V method can apply existing image pre-trained transformers as powerful backbone networks for video tasks such as video classification, spatiotemporal action detection, and video segmentation. Although we do not provide direct applications, it still has the potential to be applied to many scenarios related to video tasks. On the positive side, a powerful video understanding backbone network can improve the performance of downstream tasks and thus enhance efficiency in various scenarios, such as in the fields of smart healthcare and intelligent transportation where video understanding is required. On the other hand, if applied improperly, advanced video networks may also have negative impacts, such as being used in terrorist military activities. Researchers need to carefully consider the potential risks and impacts when applying it to real-world scenarios.

# E    LICENSE OF DATASETS AND PRE-TRAINED MODELS

All the datasets we used are commonly used datasets for academic purpose. The license of the Kinetics-400[1] is CC BY-NC 4.0[2]. The license of the Something-Something V2[3] is custom. We used the publicly available CLIP pre-trained weights provided by OpenAI[4] and the Swin Transformer pre-trained weights provided by Microsoft[5], both of which use the MIT License.

---

[1] https://www.deepmind.com/open-source/kinetics
[2] https://creativecommons.org/licenses/by/4.0
[3] https://developer.qualcomm.com/software/ai-datasets/something-something
[4] https://github.com/openai/CLIP
[5] https://github.com/microsoft/Swin-Transformer

