# OpenReview forum: "ZeroI2V: Zero-Cost Adaptation of Pre-Trained Transformers from Image to Video"
_ICLR.cc/2024/Conference — Submitted to ICLR 2024_

### Official Review · Reviewer_X4CJ · 2023-10-26

**Soundness:** 2 fair
**Presentation:** 3 good
**Contribution:** 2 fair
**Rating:** 5
**Confidence:** 4

**Summary:**

This paper introduces the ZeroI2V paradigm for adapting image models to video recognition tasks without introducing additional computation during inference. It utilizes spatial-temporal dual-headed attention (STDHA) and linear adapters to capture video dynamics and handle the domain gap between images and videos. Experimental results show that ZeroI2V achieves state-of-the-art performance while maintaining parameter and inference efficiency.

**Strengths:**

1. I think that the concept of zero-cost temporal modeling is a promising approach for image-to-video adaptation, and it makes logical sense to me as well.
2. The paper is easy to follow, and the experimental sections are well-designed.
3. The experimental results clearly demonstrate significant gains, particularly on the SSv2 dataset.

**Weaknesses:**

1. I am not fond of the term "zero-cost adaptation." In reality, the adaptation process, which involves re-training, cannot be considered zero-cost. The zero-cost aspect only applies during inference after the linear adapter has been merged with the existing weights. Referring to it as zero-cost adaptation may be an overstatement.
2. In my opinion, the full adaptation diagram still requires a significant amount of computational resources and memory for backpropagation, as there are a bunch of tunable parameters located in the shallow layers.

**Questions:**

1. Is there a training wall clock time comparison with prior works? What is the total training parameter of linear adapters used during training in Table 2/3?
2. I noticed that the best figure in Table 2 for SSv2 Top-1 is 66.3. However, in Table 4, the corresponding number is 67.7. Which setting accounts for this improvement?
3. How do you select a specific head for different $\Delta t$? Has there been any ablation study conducted?

---

> ### Author Response · Authors · 2023-11-17
>
> **Q1: Referring to it as zero-cost adaptation may be an overstatement.**
>
> We sincerely apologize for any confusion. We have repeatedly emphasized in the abstract and the main paper that our meaning of “zero-cost adaptation” is that there is no additional computational and parameter cost during inference.  We would change the name  "zero-cost adaption" to "adaption with zero extra inference cost"  or "zero-extra cost adaption" in the final version for clarity.
>
> **Q2: the full adaptation diagram still requires a significant amount of computational resources and memory for backpropagation. Is there a training wall clock time comparison with prior works?**
>
> We have supplemented the comparison of ZeroI2V and the previous PETL method regarding training time and training GPU memory usage in ***Table 14 of Appendix B***. It can be seen that although our method’s training efficiency is not as prominent as the inference efficiency compared to the previous PETL method, it is also not much more cumbersome. Compared to AIM that impose an additional computational burden for temporal modeling, our STDHA method, which does not introduce extra learnable parameters, ensures that ZeroI2V maintains superior training efficiency. And we believe that it is worthwhile and acceptable to exchange some training cost for a reduction in inference cost.
>
> **Q3: What is the total training parameter of linear adapters used during training in Table 2/3?**
>
> I presume you might be referring to Table 3/4? As in Table 2, we have already listed the number of tunable parameters. As for the total training parameters of ZeroI2V in Table 3/4, we provide the final setting that we used in Table 6 in Appendix A.1. For most cases, we use 4 adapters per ViT block (14M for ViT-B and 50M for ViT-L).
>
> **Q4: I noticed that the best figure in Table 2 for SSv2 Top-1 is 66.3. However, in Table 4, the corresponding number is 67.7. Which setting accounts for this improvement?**
>
> The result of 66.3 was obtained using a single view test (center crop), while the result of 67.7 was obtained using a multi-view test (three crops). This evaluation method is consistent with previous work.
>
> **Q5: How do you select a specific head for different Δt? Has there been any ablation study conducted?**
>
> Due to the existence of serial learnable adapters, we believe that it is unnecessary to deliberately select heads for different $\Delta t$. This is because different heads actually correspond to different feature dimensions, which in turn correspond to $W_{qkvo}$ in the attention layer. We believe that the learnable linear adapter can adaptively adjust the weights of $W_{qkvo}$ corresponding to each head during low-rank adaptation.

---

> > ### Comment · Reviewer_X4CJ · 2023-11-20
> >
> > I appreciate the authors' diligent efforts and many of my questions have been addressed.
> >
> > 1) The figure under "Param(M)" in Table 3/4 is about the inference phase rather than actual number of tunable parameters. It would be better to point this out in the caption of the table. In fact, these seem comparable or a little higher than those in previous methods in terms of the training parameters number and GPU training hours. Notably, the unmentioned work DualPath[1] surpasses this paper's performance by 2 points on ViT-Base on K400 (83.4 vs 85.4), which requires fewer FLOPs and tunable parameters. This does cast some doubt on the efficiency contributions of this paper.
> >
> > 2) Additionally, I find myself aligning with reviewer PNMk and Z2GV's perspectives on the novelty of the paper. The application of shifting operations and reparameterization techniques, while common in our community, do not appear to bring profound insights to the I2V area.
> >
> > Considering these observations, I view this paper as borderline and eagerly await other reviewers' opinions for further evaluation.
> >
> > [1] Jungin Park, Jiyoung Lee, and Kwanghoon Sohn. Dual-path adaptation from image to video transformers. In Proceedings of the IEEE/CVF Conference on Computer Vision and Pattern Recognition (CVPR), pages 2203–2213, 2023.

---

> ### Author Response · Authors · 2023-11-21
> **Further explanation for Reviewer X4CJ**
>
> Thank you for your valuable suggestion. Here we provide further explanation for your existing concerns:
>
> 1. **Comparsion with DualPath**:  We have also noticed this work. DualPath achieves high computational efficiency by using fewer sampling frames in the spatial adaptation branch. In the temporal adaptation branch, it uses more sampling frames, and after reducing the resolution of each frame, it stitches them into a grid-like frameset to control the computation.
>
>    - In terms of technical contributions, **our method and DualPath’s method are complementary.** We simply followed the settings of previous works like ST-Adapter and AIM. In fact, we can also replace DualPath’s temporal adaptation branch with our temporal modeling, or enhance the temporal modeling capability of the DualPath’s spatial adaptation branch without increasing additional computation. Our linear adaptation strategy can also reduce the parameter and computational load of DualPath during inference.
>    - In terms of performance, we believe that the temporal modeling capability of our STDHA actually surpasses the “Grid attention” used by DualPath. This can be seen from the comparison with SIFAR [1] in ***Table 13 of Appendix B*** (SIFAR uses the same temporal modeling method as DualPath).
>    - We actually tried to reproduce the performance of their method using the code provided by the DualPath authors, but we failed. The authors did not provide any further replies or explanations, nor did they provide model checkpoints. Our own implementation also **failed to reproduce the performance**, we also notice similar reproduction issue is also encountered by others in the GitHub issues.
>
>    Therefore, we did not compare with DualPath in the origin paper. We can supplement this in later versions.
>
> 2. **The contribution of our method:** We agree with your statement that the application of shifting operations and reparameterization techniques is common in our community. However, our work is not a simple stack of these techniques:
>    - The shift operation is not the core contribution of our STDHA. We just use it to provide an efficient implementation of STDHA (for details, please refer to **our answers to Q3 of Reviewer PNMk and Reviewer Z2GV**);
>    - The reparameterization technique is also just a means for us to implement linear adaption. Our contributions lie in our full adaption strategy and our comprehensive experiments.
>
> [1] Fan, Quanfu, Chun-Fu Chen, and Rameswar Panda. "Can an Image Classifier Suffice For Action Recognition?." *International Conference on Learning Representations*. 2021.

---

> ### Author Response · Authors · 2023-11-23
> **Looking forward to your feedback**
>
> Thanks for your constructive suggestions again. We value your comments and have made efforts to revise the paper. Please kindly let us know if our response addressed your concerns. We are willing to respond to any further questions before the rebuttal ends **(within 6 hours)** :-)

---

### Official Review · Reviewer_Z2GV · 2023-10-31

**Soundness:** 3 good
**Presentation:** 2 fair
**Contribution:** 2 fair
**Rating:** 5
**Confidence:** 3

**Summary:**

The authors introduce Zero2IV, a method to adapt image models to the video domain that avoids full fine-tuning and does not increase computational cost during inference. Two aspects of the problem are dealt with: temporal modeling and the image-to-video domain gap. The first is addressed by Spatio-Temporal Dual-Headed Attention (STDHA). In STDHA, some heads of the transformer model are assigned to model temporal relations, while the other heads model spatial relations. The second is addressed by densely placed linear adapters. The computational cost of the original image model is kept the same during inference on video inputs by re-parameterizing the model post-training.

**Strengths:**

- The results of ZeroI2V in Section 4.3 show that it has consistent advantages over previous PETL methods in terms of accuracy when the inference efficiency of the methods is taken into account.
- The ablation studies on the hyperparameter settings of ZeroI2V show that the proposed components each contribute positively to its performance.

**Weaknesses:**

- My main concern regards the trainability of ZeroI2V: I did not find the training time and training GPU memory usage of ZeroI2V mentioned in the main paper. Since there are linear adapters densely placed throughout the network, this makes it unclear whether ZeroI2V is much more cumbersome to train than previous PETL methods.
- The authors claim ZeroI2V is a general method to adapt image models to the video domain, but the experiments are only done on the action recognition task, which does not require fine-grained spatial understanding as opposed to tasks like video segmentation. In order to properly support this claim there need to be experiments on another video task.

**Questions:**

- Can you clarify the novelty of ZeroI2V compared to the Patch Shift Transformer introduced in [1]?
- What is the difference between head relocation QKV and head relocation KV (STDHA) in Table 1a?
- Which configuration/hyperparameter setting is chosen for ZeroI2V, based on the ablations, for the experiments in Section 4.3 that compare it to the state of the art?
- Is the channel change ratio $R_{c}$ supposed to be the ratio $k:h-k$?
- In Table 5, why is ST-Adapter missing? It seems most similar to ZeroI2V in terms of efficiency and accuracy on K400 and SSv2.

(writing-related:)
- What is the benchmark mentioned on page 2 in “establish a benchmark” on page 2? Usually this means there is a new dataset introduced.
- How is “offering powerful capabilities” a "key perspective" of temporal modeling? It's unclear what idea the authors are trying to get across.
- Saying ZeroI2V is “reducing the difficulty” of image-to-video adaptation is also vague. It would be better to specifically mention reducing the inference cost.

Typos:
- Page 1 paragraph 1 sentence 2: Missing “and” between “CLIP” and “DINO”
- Page 1 paragraph 1 sentence 4: Remove “the” from before “parameter efficient transfer learning”.
- Page 2 paragraph 2 sentence 2: Why is the word “images” treated as a proper noun?
- Page 2 paragraph 4 sentence 5: Incomplete sentence. Remove “find” from beginning of sentence.
- Page 3 paragraph 2 sentence 3: Replace “image” with the plural “images”
- Page 3 paragraph 3 sentence 1: Use past tense “was” instead of “is”. In sentence 3, missing definite article “the” before “video domain”.
- Last sentence on page 3: Remove “then”. Use a period after “details” and begin a new sentence.
- Page 4 paragraph 2 sentence 4: Typo in word “difficulty”.
- Page 4 paragraph 4 sentence 1: Incomplete sentence. Replace “given an input” with “the input is a ...“
- Page 4 paragraph 4 sentence 2: Should the groups be of size h-k and k (instead of n-k and k)?
- Page 5 paragraph 4 sentence 1: Use “Assume“ instead of “assuming”.
- Table 1 caption add a space between “section” and “is”.

[1] Xiang et al. "Spatiotemporal Self-attention Modeling with Temporal Patch Shift for Action Recognition." ECCV, 2022.

---

> ### Author Response · Authors · 2023-11-17
>
> **Q1: Trainability of ZeroI2V**
>
> We have supplemented the comparison of ZeroI2V and the previous PETL method regarding training time and training GPU memory usage in ***Table 14 of Appendix B***. It can be seen that although our method’s training efficiency is not as prominent as the inference efficiency compared to the previous PETL method, it is also not much more cumbersome. Compared to AIM that impose an additional computational burden for temporal modeling, our STDHA method, which does not introduce extra learnable parameters, ensures that ZeroI2V maintains superior training efficiency. We believe that it is worthwhile and acceptable to exchange some training cost for a reduction in inference cost.
>
> **Q2: There need to be experiments on another video task**
>
> Thank you for your valuable suggestions. To understand the capability of our method in fine-grained spatial understandingWe evaluate our method on action detection dataset AVA. Following the setting of VideoMAE. As shown in ***Table 9 of Appendix B*** , compared to using the original image CLIP features, our ZeroI2V achieved a significant performance improvement (26.4 vs 18.3) with the same number of parameters and computation. It’s noteworthy that our method was not pre-trained on action recognition datasets such as Kinetics. Instead, we directly applied image-to-video transfer on the AVA dataset. Remarkably, our method still managed to achieve performance on par with full-finetuning methods and self-supervised methods that underwent pre-training using the Kinetics dataset, even when using only 8 frames as input. In summary, our ZeroI2V demonstrates outstanding potential in video tasks beyond recognition.
>
> **Q3: Can you clarify the novelty of ZeroI2V compared to the Patch Shift Transformer introduced?**
>
> The TPS operation in the Patch Shift Transformer is placed before the self-attention layer. It achieves efficient modeling by mixing information from frames at different times, which is more like an extension of the channel shift proposed in TSM, rather than a profound improvement on the original MHSA operation. Our STDHA, on the other hand, starts from the redundancy of heads in MHSA. By assigning some heads that originally performed spatial modeling to complete temporal modeling tasks, we believe it maintains the purity of information within the same head (in contrast, in TPS, each head contains a portion of channels from other frames when calculating attention, i.e., all heads are simultaneously performing spatio-temporal modeling). This allows us to ultimately achieve better spatio-temporal modeling capabilities. This point is also demonstrated in Table 13 of Appendix B, where under the same settings, Swin-B w/ STDHA achieved significantly better performance than PST-B (70.0 vs 67.4).
>
> Additionally, we understand that the reason you may find this confusing is because we use inter-frame attention for temporal modeling in STDHA, which also requires us to use a shift operation in our implementation. However, we believe that the key contribution of our proposed STDHA is not just in our use of inter-frame attention for temporal modeling. Rather, it lies in demonstrating the effectiveness and efficiency of using different heads for simultaneous spatial and temporal modeling, and in providing an efficient implementation based on inter-frame attention. In the future, we will explore more temporal modeling operations for the temporal head.
>
> **Q4: What is the difference between head relocation QKV and head relocation KV (STDHA) in Table 1a?**
>
> In the temporal head, when calculating attention, “head relocation QKV” means that we use the QKV corresponding to non-current moment frames for attention calculation. On the other hand, “head relocation KV (STDHA)” means that we use the Q of the current moment frame and the KV corresponding to non-current moment frames for attention calculation, thereby implementing inter-frame attention.
>
> **Q5: Which configuration/hyperparameter setting is chosen for ZeroI2V**
>
> In Appendix A.1, Table 6, we provide the final setting that we used in Section 4.3.
>
> **Q6: Is the channel change ratio R_c supposed to be the ratio k:ℎ−k?**
>
> $R_c$ represents the ratio of the number of channels involved in temporal modeling to all channels. Specifically, $R_c$ denotes $k:h$ in STDHA.
>
> **Q7: In Table 5, why is ST-Adapter missing?**
>
> We have already shown the latency when using two ST-Adapters in Table 2, which is significantly higher than ZeroI2V (38.8 vs 28.9). ST-Adapter uses a different number of adapters for K400 and SSv2, so we need two lines to present it. Due to space constraints, we did not include it in the comparison in Table 5. We will update it in our paper after adjusting the layout.

---

> ### Author Response · Authors · 2023-11-17
> **Explanation on writing-related questions.**
>
> Thank you for your suggestions regarding grammar. We have corrected these issues in the revised paper. Additionally, we would like to address your questions about the content of the writing here.
>
> > **What is the benchmark mentioned on page 2 in “establish a benchmark” on page 2?**
>
> We agree with your statement that a benchmark typically introduces a new dataset. In this case, we use the existing Something Something v2 dataset as our dataset for evaluating the temporal modeling capabilities of our model after transfer. We compared common image-to-video adaptation methods under fair comparison conditions, so we refer to it as “establishing a benchmark”. Similar expressions are also used in the introduction of ST-Adapter.
>
> > **How is “offering powerful capabilities” a "key perspective" of temporal modeling?**
>
> What we wanted to express here is that the “Capability of capturing the temporal dynamics” is a “key perspective” of temporal modeling. We have corrected this in the revised paper.
>
> > **Saying ZeroI2V is “reducing the difficulty” of image-to-video adaptation is also vague**
>
> Here, we are not just aiming to reduce the inference cost, but we are also concerned with reducing the training cost in image-to-video adaptation. STDHA, by not introducing additional learnable parameters and retaining the original spatial head for image modeling, also reduces the difficulty of training to a certain extent.

---

> ### Author Response · Authors · 2023-11-23
> **Looking forward to your feedback**
>
> Thanks for your constructive suggestions again. We value your comments and have made efforts to revise the paper. Please kindly let us know if our response addressed your concerns. We are willing to respond to any further questions before the rebuttal ends **(within 6 hours)** :-)

---

> ### Comment · Reviewer_Z2GV · 2023-11-23
>
> Thank you for your comprehensive response to my questions. It has improved my understanding of of ZeroI2V's implementation. Additionally, the efficacy of ZeroI2V on the action detection task shows that it may be valuable for more dense prediction tasks, although action detection does not require as much pixel-wise spatial understanding in videos compared to video object detection or object segmentation. My remaining concerns are the following:
>
> 1&5. Regarding the trainability of the method, it is concerning that different hyperparameters are chosen for different datasets. This suggests that a careful choice of hyperparameters is necessary to achieve good performance in practice, which might make the method cumbersome to use. When it comes to the presentation, it is misleading to put the term "zero-cost" in the paper when the training time is increased as it is compared to the state of the art.
>
> 3. It is still a bit difficult to understand how large of a difference the implementation of ZeroI2V is compared to previous methods. A figure showing how tokens are used in the attention operations in ZeroI2V vs previous methods like TPS would probably be helpful in clarifying the novelty.
>
> 7. Unless I'm mistaken ST-Adapter is missing from Table 5 in the most recent version, perhaps this can be updated in the final version?
>
> The concern around trainability and ease of use in practice is the main point that leads me to keep my rating.

---

> > ### Author Response · Authors · 2023-11-23
> >
> > **Q: different hyperparameters are chosen for different datasets. which might make the method cumbersome to use.**
> >
> > In fact, we did not make many adjustments to the hyperparameters. Our training settings are mostly based on AIM. Moreover, previous works such as EVL, AIM, and ST-Adapter have used slightly different settings for different datasets to achieve better performance. For instance, EVL used a deeper decoder for the SSv2 dataset, while AIM and ST-Adapter used a larger number of Adapters for the SSv2 dataset. We believe that our method is not more cumbersome than previous works in this regard.
> >
> > **Q: some writing-related suggestions.**
> >
> > Thank you for your valuable feedback. We will revise the content of our paper in the final version according to your suggestions, making our methods and experimental results easier to understand.

---

### Official Review · Reviewer_PNMk · 2023-11-01

**Soundness:** 3 good
**Presentation:** 3 good
**Contribution:** 3 good
**Rating:** 6
**Confidence:** 3

**Summary:**

The paper introduces zeroI2V, an video model understanding model based on the pre-trained image models. The authors propose an STDHA which performs spatio-temporal modeling at no additional cost at inference time. The action recognition results on SSV2 and K400 are solid and convincing.

**Strengths:**

- The paper is clearly written, easy to follow.
- the introduced STDHA works as expected and the results are comprehensive and convincing.

**Weaknesses:**

- The results are convincing but needs a bit more illustration. For example MViTv2-L works better on SSv2, is that from the model design or fully supervised training or something else? Same thing for the AIM better on the K400, where does the performance gap comes from, the design or something else? what are the advantage and disadvantages of the proposed STDHA comparing against the commonly used action recognition models (not only the fine-tuned CLIP models).

- The novelty is a bit limited or not well highlighted, as the inter-frame attention is not originally from this paper. The paper still has a solid idea on adapters at no additional cost, i would encourage the authors to give the intuition a bit more illustration, e.g. why pick the inter frame attention as part of the proposed STDHA.

- The Vis. are not clear enough, consider to put a CLIP VIT activation map there for comparison.

**Questions:**

- See my first comment in the weakness.

---

> ### Author Response · Authors · 2023-11-17
>
> **Q1: The results are convincing but needs a bit more illustration**
>
> Thank you for your valuable suggestions. Due to space constraints, we did not elaborate in detail on the comparison of results. We will supplement it here and update it in our paper after adjusting the layout:
>
> - **Why MViTv2-L works better on SSv2？** Firstly, from the perspective of model design, MViTv2 introduces more efficient downsampling designs compared to the original ViT, thereby resulting in lower FLOPs. This allows it to use a higher spatial resolution (312) and sampling frame number (40). Secondly, our method is based on image pre-training, while MViTv2 is based on video weights pre-trained on K400. As SSv2 places more emphasis on temporal modeling capabilities, MViTv2-L can achieve a better performance trade-off.
> - **Why AIM better on the K400？** We understand that the K400 dataset places more emphasis on spatial modeling capabilities. Our STDHA, by making some heads that originally performed spatial modeling complete temporal modeling, has to some extent reduced the model’s spatial modeling capabilities. On the other hand, AIM, through the stack of temporal attention and spatial attention, fully retains the model’s original spatial modeling capabilities. Therefore, it achieves slightly stronger performance on K400, but at the cost of nearly 30% additional computational overhead.
>
> **Q2: The advantage and disadvantages of the proposed STDHA comparing against the commonly used action recognition models**
>
> - **advantage**：The first obvious advantage is that STDHA is highly efficient, requiring no additional learnable parameters or computational overhead. In addition, compared to other efficient temporal modeling methods, our STDHA is specifically designed for the characteristics of the attention mechanism and has stronger spatio-temporal modeling capabilities (we have discussed this in Section 3.2 of the main paper). Finally, due to the popularity of the attention mechanism, our method has a very wide range of applications and is plug-and-play (we have shown in Table 13 of Appendix B that our method can also achieve good performance when using Swin Transformer as the backbone).
> - **disadvantage**：The main drawback of STDHA is that it makes some heads, which originally performed spatial modeling, complete temporal modeling. This to some extent impairs the model’s spatial modeling capabilities. This may result in slightly poorer performance in scenarios that require high spatial information modeling (for example, its performance on K400 is slightly inferior to AIM).
>
> **Q3: Why pick the inter frame attention as part of the proposed STDHA:**
>
> - **Motivation of the choice of inter-frame attention:** Our method is designed for image-to-video adaptation, so we start from the perspective of trying to reduce the difficulty of image model transfer. A direct idea is to tile video frames into images for spatio-temporal modeling, but this is not ideal in terms of both computational efficiency and performance. Then we considered inter-frame attention. Since only the key and value of the current frame are replaced with those of other frames, it can be assumed that the data distribution of the key and value has not changed much when calculating attention. From this perspective, we replaced the temporal attention in AIM with inter-frame attention. We found that it has stronger temporal modeling capabilities in the image-to-video scenario:
>
> | Method         | Spatial Modeling      | Temporal Modeling      | FLOPs | SSv2 Top1 |
> | -------------- | --------------------- | ---------------------- | ----- | --------- |
> | AIM       | spatial attention     | temporal attention     | 624   | 66.4      |
> |                | spatial attention     | inter-frame attention  | 640   | 68.4      |
> | ZeroI2V (ours) | spatial head in STDHA | temporal head in STDHA | 422   | 67.7      |
>
> Moreover, the implementation of inter-frame attention is very simple and has the same computational cost as the original spatial attention, we can implement STDHA in a highly efficient manner.
>
> - **The inter-frame attention is not originally from this paper：** We believe that the key contribution of our proposed STDHA is not just in our use of inter-frame attention for temporal modeling. Rather, it lies in demonstrating the effectiveness and efficiency of using different heads for simultaneous spatial and temporal modeling, and in providing an efficient implementation based on inter-frame attention.
>
> **Q4: CLIP VIT activation map there for comparison.**
>
> Thank you for your valuable suggestions. We have added the CLIP VIT activation map for a more intuitive comparison in ***Figure 3 of Appendix C*** in our revised paper. It can be seen that STDHA pays more attention to the interaction of objects in the video (such as the touch between hands and cups or Rubik’s cubes), while CLIP only focuses on individual objects and does not capture the spatio-temporal dynamics in the video well.

---

> ### Author Response · Authors · 2023-11-23
> **Looking forward to your feedback**
>
> Thanks for your constructive suggestions again. We value your comments and have made efforts to revise the paper. Please kindly let us know if our response addressed your concerns. We are willing to respond to any further questions before the rebuttal ends **(within 6 hours)** :-)

---

### Official Review · Reviewer_HQJm · 2023-11-02

**Soundness:** 3 good
**Presentation:** 3 good
**Contribution:** 3 good
**Rating:** 8
**Confidence:** 4

**Summary:**

This paper focuses on adapting pre-trained image transformer to video transformer efficiently. Two main techniques are proposed. One is to split the pre-trained self-attention heads into spatial and temporal heads, where temporal heads are doing self-attention across frames to learn temporal information. The second technique is to use linear adapters to tune the frozen model. Then after training, these adapters could be fused into the backbone, without introducing new parameters/computations. The method achieves competitive performance with previous efficient adaptation works on multiple datasets.

**Strengths:**

1.	The idea to split the pre-trained self-attention heads into spatial and temporal heads is interesting. And it is reasonable since there are redundancies in the pre-trained ViT.
2.	Using linear adapters to tune the model and fuse it with the backbone later is also technically sound.
3.	The proposed method achieves competitive performance on multiple video datasets with previous works, without increasing parameters and FLOPs.
4.	The paper is well written and easy to follow

**Weaknesses:**

1.	I am curious about how is the STDHA implemented. Because it needs to split the heads into spatial and temporal, I am assuming it will introduce some other operations, although they may not contribute to FLOPs, but may still slow down the latency. However, in Table 2, the proposed method has exactly the same latency as the baseline.
2.	In Table 1 (b), what is the meaning of 1/2 head?
3.	In Table 4, ST-Adapter ViT-L/14 has a performance of 72.3/93.9, which is higher than the proposed method. I think it would be better to show the full comparison, and I don’t think it will degrade the significance of the work.
4.	UCF101 and HMDB51 are very similar to K400. It could better show the effectiveness of the proposed method to show some results on other datasets such as Diving48, Epic-kitchens, etc.

**Questions:**

Please see the weakness part

---

> ### Author Response · Authors · 2023-11-17
>
> **Q1: how is the STDHA implemented?**
>
> Our choice of inter-frame attention for temporal modeling in STDHA allows us to directly apply the shift operation to the key and value tensors associated with the temporal head. Shift operation is very lightweight and can be almost ignored compared to other operations that are counted into FLOPs (such as matrix multiplication). Specifically, we directly modified the `nn.MultiHeadAttention` in PyTorch. It’s worth mentioning that our STDHA does not conflict with some commonly used efficient attention implementations such as FlashAttention, and we can combine them to achieve higher computational efficiency.
>
> **Q2:  In Table 1 (b), what is the meaning of 1/2 head?**
>
> $\Delta t=1/2$ means that half of the channels in this temporal head have the time offset of 1 and the other half have -1. This experiment actually violates our original intention of keeping the purity of the head, so we just treat it as an extreme case of ablation experiments.
>
> **Q3: It would be better to show the full comparison with ST-Adapter**
>
> Thank you for your valuable suggestion. Due to space constraints, we choose one method for comparison at different model sizes. We will add a more comprehensive comparison after adjusting the layout in the future.
>
> **Q4: It could better show the effectiveness of the proposed method to show some results on other datasets such as Diving48, Epic-kitchens**
>
> We have already demonstrated on the SSv2 dataset that ZeroI2V can still achieve good performance even when there is a large distribution difference with the k400 dataset. Additionally, following your suggestion, we have supplemented the experimental results on Diving48. We achieved a top-1 accuracy of **91.4%**, significantly surpassing AIM (90.6%). This further showcases the robust performance of our method on small datasets. Detailed results and comparisons can be found in ***Table 11 of Appendix B*** in our revised paper.

---

### Author Response · Authors · 2023-11-22
**We are looking forward to hearing from you about any further feedback.**

Dear reviewers,

We thank all reviewers for the valuable and constructive comments. We would like to provide a summary of the comments and discussions from the reviewers:

- All reviewers agreed that our work presents comprehensive and convincing experimental results, and easy to follow.

- Reviewer HQJm suggested that our method needs to be tested on a small-scale dataset that significantly differs from K400. Therefore, we have supplemented the experimental results on the Diving48 dataset in ***Table 11 of Appendix B***, and achieved excellent performance.

- Reviewer PNMk suggested that our method needs to be validated on video tasks beyond the action recognition task, especially those requiring fine-grained spatial understanding. Therefore, we have supplemented the experimental results on the action detection dataset AVA in ***Table 9 of Appendix B***, and achieved persuasive performance.

- Reviewer HQJm approved the technical contributions and soundness of our method. The other reviewers had some doubts about the motivation and novelty of our method. We have provided detailed explanations for this in our ***answers to Q3 of Reviewer PNMk and Reviewer Z2GV***.

- Reviewers Z2GV and X4CJ raised questions about the training cost of our method. We have supplemented the comparison of ZeroI2V and the previous PETL method regarding training time and training GPU memory usage in ***Table 14 of Appendix B***. The results show that the training cost of our method is comparable or a little higher than those in previous methods. Reviewer X4CJ also acknowledged this point.

We are looking forward to hearing from you about any further feedback.

Best, Authors

---

### Meta-Review · Area_Chair_fLfb · 2023-12-03

**Metareview:**

This submission has elicited a range of evaluations from the reviewers, with scores of 6, 8, 5, and 5. This scoring reflects a division in reviewer opinions, with two advocating for acceptance and two suggesting rejection.

"ZeroI2V" is presented as an interesting method for adapting image models to video recognition tasks. The key contributions include the introduction of Spatial-Temporal Dual-Headed Attention (STDHA) and linear adapters. These elements are designed to effectively capture video dynamics and bridge the gap between image and video domains, all while avoiding additional computational burdens during inference. The authors assert that their method achieves state-of-the-art performance and maintains efficiency in both parameters and inference.

However, the manuscript faces critical scrutiny regarding its novelty, the extent of its comparative analysis with existing methods, and practical considerations related to training efficiency. Reviewers have raised important concerns about the need for dataset-specific hyperparameter tuning, which could potentially undermine the practicality and broader applicability of the proposed method. Furthermore, the term "Zero-Cost" used in the manuscript can be misleading, as additional training time is still required.

In light of these considerations, and after a thorough evaluation of the manuscript's strengths (Why Not Lower) and weaknesses (Why Not Higher) below, I conclude that the submission does not currently meet the threshold for acceptance. It shows promise and introduces intriguing concepts, but key issues related to novelty, comparative analysis, and practical training aspects need to be more robustly addressed. Consequently, I recommend rejection at this stage, while encouraging the authors to refine and improve their approach, taking into account the detailed feedback provided, for future submissions.

**Justification For Why Not Higher Score:**

- Limited Novelty and Contribution: 3 reviewers have expressed concerns about the novelty of the method, highlighting that the shift operations and reparameterization techniques employed are common in the community. The manuscript doesn't seem to offer profound insights into the Image-to-Video (I2V) adaptation area.
- Training Cost Concerns: a reviewer noted that ZeroI2V, despite its inference efficiency, might still be cumbersome in practice due to different hyperparameters needed for different datasets. This undercuts the "zero-cost" adaptation claim, as the adaptation process involves re-training, which is not cost-free.
- Comparative Analysis with Other Works: a reviewer pointed out that the manuscript lacks a convincing demonstration of ZeroI2V's superiority over existing methods like DualPath, which surpasses ZeroI2V in certain aspects. This casts doubt on the efficiency and performance contributions claimed by the authors.
- Unclear Implementation and Theoretical Foundations: The manuscript could benefit from more detailed explanations and visualizations of how ZeroI2V's attention operations work, especially in comparison with previous methods like TPS.

**Justification For Why Not Lower Score:**

- Good Experimental Results: The method demonstrates competitive performance on multiple video datasets. The ablation studies and experiments, such as those on the Diving48 and AVA datasets, showcase the robustness and potential of ZeroI2V in various tasks.
- Technical Soundness: Despite some concerns, the technical approach makes sense. The idea of splitting pre-trained self-attention heads into spatial and temporal heads, along with using linear adapters for tuning the model, makes sense and considered to be well-grounded.
- Good Presentation and Clarity: The manuscript is well-written and easy to follow, which was acknowledged by most reviewers. The authors have also responded comprehensively to the reviewers' comments, showing diligence in addressing concerns.
- Positive Aspects of the Method: The concept of zero-cost temporal modeling for image-to-video adaptation is innovative and promises efficiency in practical applications. The use of STDHA and linear adapters is a novel approach in the context of video recognition tasks.

---

### Decision · Program_Chairs · 2024-01-16

Reject